# Digital Twins in Healthcare: Is It the Beginning of a New Era of Evidence-Based Medicine? A Critical Review

**DOI:** 10.3390/jpm12081255

**Published:** 2022-07-30

**Authors:** Patrizio Armeni, Irem Polat, Leonardo Maria De Rossi, Lorenzo Diaferia, Severino Meregalli, Anna Gatti

**Affiliations:** 1LIFT Lab and CERGAS, GHNP Division and Claudio Demattè Research Division, SDA Bocconi School of Management, 20136 Milano, Italy; 2LIFT Lab, Claudio Demattè Research Division and GHNP Division, SDA Bocconi School of Management, 20136 Milano, Italy; irem.polat@sdabocconi.it (I.P.); anna_gatti@sdabocconi.it (A.G.); 3SDA Bocconi School of Management, 20136 Milano, Italy; leonardo.derossi@sdabocconi.it (L.M.D.R.); lorenzo.diaferia@sdabocconi.it (L.D.); severino.meregalli@sdabocconi.it (S.M.)

**Keywords:** digital twins, precision medicine, digital technologies, convergence, personal health management, clinical trials design, hospital operations

## Abstract

Digital Twins (DTs) are used in many different industries (e.g., manufacturing, construction, automotive, and aerospace), and there is an initial trend of applications in healthcare, mainly focusing on precision medicine. If their potential is fully unfolded, DTs will facilitate the as-yet-unrealized potential of connected care and alter the way lifestyle, health, wellness, and chronic disease will be managed in the future. To date, however, due to technical, regulatory and ethical roadblocks, there is no consensus as to what extent DTs in healthcare can introduce revolutionary applications in the next decade. In this review, we present the current applications of DTs covering multiple areas of healthcare (precision medicine, clinical trial design, and hospital operations) to identify the opportunities and the barriers that foster or hinder their larger and faster diffusion. Finally, we discuss the current findings, opportunities and barriers, and provide recommendations to facilitate the continuous development of DTs application in healthcare.

## 1. Introduction

In the last two decades, the advent of the Internet of Things (IoT) and the diffusion of technologies such as sensors and connected actuators are changing the way data are exchanged among different sources, generating a growing production of big data. Today, scientific advances in big data analytics, cloud computing, and the integration of artificial intelligence (AI) allow for the storing and processing of IoT data, and this is the ground for the potentially large diffusion of one of the most interesting advancements in the field of technology, Digital Twins (DTs). At its root, the concept of DTs relies on creating virtual alter-egos of objects, living organisms, spaces, or processes.

DTs take their origins from mirrored systems, or simulated environments, created by NASA in the 1970s to monitor unreachable physical spaces (e.g., spacecraft in mission) [1]. The most famous early application of DTs occurred during the Apollo 13 mission when one of the oxygen tanks exploded two days after launch, and engineers on the NASA flight control team in Houston had to model and test possible solutions in a simulated environment. They guided astronauts to build an improvised air purifier with materials available in the spacecraft to get the Apollo 13 crew back to earth [1]. This example is considered the precursor of DTs, allowing the connection between physical and virtual spaces even before the Digital Twin (DT) concept gained popularity [2]. However, the earliest instances do not fully represent proper DTs, since seamless connection and real-time data exchange remained insufficient to create a continuous digital twinning to the physical world [1].

In 2002, Michael Grieves introduced the initial conceptual model of DTs during his presentation on product life-cycle management for manufacturing. Since then, the use of DTs has expanded through very different application fields: for example, according to NASA (2012) [3], ‘‘A DT is an integrated multiphysics, multiscale, probabilistic simulation of an as-built vehicle or system that uses the best available physical models, sensor updates, fleet history, etc., to mirror the life of its corresponding flying twin.’’ [4]. In 2014, Grieves provided a general formalization of the DT concept and identified three major elements of DTs [5]: (1) A real space representing a physical object (e.g., an object, a process, a person, a phenomenon); (2) A virtual space containing a virtual object; and (3) A digital thread connecting the real space and virtual space (allowing the data to flow from real space to virtual space and the information to flow from virtual space to real space) [1,2]. The key idea is to computationally model those systems to develop and test them more quickly, economically, and with less potential negative consequences than in real life. In the last two decades, the DT concept has been applied in many manufacturing and process-related contexts to predict potential system failures earlier [6]. 

DTs have attracted research and business interest in recent years [2]. For instance, DT appears as the second most discussed technology application within the Digital Health trend (the mHealth application having the first place) in the LIFT Radar 2021 [7], where each year technology applications at the convergence of digital and life sciences are assessed. Additionally, DT is the third trending technology for 2020 according to the IEEE Computer Society, [8] where technology experts reveal their annual predictions for the most widely adopted future technology trends. Finally, in August 2020, Gartner, one of the leading research organizations in the world, pointed out “digital me” (digital models representing humans in both real and virtual worlds) as a must-know technology that will significantly affect society, health, and business during the next decade. However, despite the high interest in DT applications, their current use remains well below its potential, particularly in life sciences domains [1]. Until now, commercially available applications have not been able to offer a precise and cost-effective tool that replicates an individual’s complete genetics, chemistry, anatomy, lifestyle, and medical history over time [7]. Yet, an accurate DT development requires a broad understanding of each part within a system, the relationship among these parts, and the analytic power to evaluate the effect of variables introduced to the system. 

Currently, DTs are used in manufacturing and construction, automotive, and aerospace industries, including product design and service management, product life prediction, and real-time monitoring of equipment [4], and there is an initial trend of applications in healthcare. DTs hold great potential especially in precision medicine, where DTs can be used to simulate individual therapies and visualize potential therapy results and disease progression for each patient [9]. The interest is growing thanks to proof of concept in other industries and the increasing availability of technological devices to collect patient data (e.g., wearables). Moreover, there are already successful implementations of DTs in healthcare for predictive maintenance and performance optimization of medical devices and hospital management systems. Since the literature for DT applications in healthcare is relatively new compared to other fields, this paper aimed at (i) reviewing the current applications of DTs in healthcare, (ii) identifying the opportunities and the barriers that hinder a larger and faster diffusion of DTs in healthcare and (iii) providing a set of recommendations to companies and policymakers to facilitate the growth of DTs application in healthcare. 

The rest of the paper is organized as follows. Section 2 introduces the concept and the current applications of DTs in healthcare. First, key elements and critical enabling technologies for DTs are presented to have a deeper understanding on the convergence among digital technologies, DTs and the healthcare field. Second, current applications of DTs in healthcare are classified and assessed under three main goals of use cases: precision medicine, clinical trials, and hospital management. In Section 3, open issues and limitations are presented. The discussion and ideas for future work are provided in Section 4. Finally, the conclusion of the work is presented in Section 5.

## 2. Concept and Current Applications of DTs in Healthcare

### 2.1. Concept of DTs in Healthcare

In healthcare, a DT is a virtual copy of a physical object or process, such as a patient, their anatomical structure, or a hospital environment. Currently, DTs in healthcare propose to dynamically reflect data sources such as electronic health records (EHRs), disease registries, “-omics” data (e.g., genomics, biomics, proteomics, or metabolomics data) as well as physical markers, demographic, and lifestyle data over time of an individual [10,11]. Thanks to the evolution of underlying technologies (e.g., IoT, AI) and increasingly diverse, accurate, and accessible data (e.g., biometric, behavioral, emotional, cognitive, and psychological data), research and potential applications of DTs in healthcare have seen increasing interest [6].

If their potential is fully unfolded, DTs will facilitate the as-yet-unrealized potential of connected care and alter the way lifestyle, health, wellness, and chronic disease will be managed in the future. Accordingly, DTs could be “fed” with diverse and real-time information obtained by wearables and other sources of self-reported data, e.g., from mobile health applications. Care providers could access a patient’s DT to see personalized information far beyond what is currently available while making treatment decisions or providing recommendations (Figure 1).

Drawing on Grieve’s definition [5] of DT:

**The physical object** can be a patient, a medical device, a wearable device, an external factor (e.g., social behavior, weather, air quality, or even government policies influencing patient health), or a system consisting of these groups of objects (e.g., a hospital).

**The virtual object** is the medical device model, wearable device model, digital person model, external factor model, and digital system models. 

**The digital thread** is healthcare data, including real-time data detected from medical and wearable devices or external factors, simulation data from digital models, historical health data and electronic health records (EHRs) from healthcare institutions, and service data from platforms that enable the communication between the physical and virtual objects and spaces [4].

Y. Liu et al. (2019) [4] describe the operation mechanism of DTs in healthcare with the following stages. First, DT models must be built on a physical object using advanced modeling techniques and tools (e.g., SysML, Modelica, SolidWorks, 3DMAX, and AutoCAD). Second, real-time data connection and exchange between physical and virtual objects should be executed through health IoT and mobile internet technologies. Third, simulation models are tested and validated by quick execution and calibration. Fourth, models are continuously adjusted accordingly to optimize and iterate DT models. Finally, following the behavior of the virtual twin, model results (e.g., diagnosis results) are sent back to the patients. With respect to the DT application on patients, the ultimate vision is to have a lifelong, personalized patient model that is updated with each measurement, scan, or exam, including behavioral and genetic data.

#### 2.1.1. Data Generation for DT Implementation

Building a DT requires sensors and other electronic components to sense and gather information from a physical asset. According to Schwartz et al. (2020) and Voigt et al. (2021), an automobile is equipped with more than 50 sensors and minicomputers that continuously monitor its functioning, which allows for online equipment monitoring, increasing flexibility and personalized services [6,9]. However, in the healthcare field, technical, ethical, and financial constraints hinder the data acquisition, while instead, the full development of human DT requires an extensive amount of data combining individual and population level representations to optimally support the clinical decision-making process [12]. 

Recent advancements in Digital Health technologies foster the collection of several clinical and para-clinical data with digital biomarkers that allows for day-to-day passive patient monitoring. Digital biomarkers are quantifiable, measurable health indicators that use devices such as smartphones, wearable, implantable (e.g., implantable cardio defibrillators) and ingestible (e.g., smart pills) sensors to collect and measure the biological (e.g., blood glucose), physiological (e.g., heart rate, blood pressure), or behavioral parameters [6,9]. In addition to digital biomarkers, DT models also require biomarkers not measured directly or requiring invasive procedures. For instance, in the domain of precision cardiology (e.g., for coronary artery disease, aortic aneurysm, valve prostheses, and stent design), the combination of cardiovascular imaging and computational fluid dynamics enables non-invasive flow field characterization and the diagnostic metrics calculation [12].

In summary, human DTs can benefit from the health sources where each living and non-living entity generate a massive digital footprint: (i) a formal healthcare system (e.g., electronic health records, lab test results, medical images, insurance, and pharmacy data); (ii) digital health devices (e.g., Bluetooth-connected glucose monitors, smart watches); (iii) patient surveys; (iv) real-world health data sources obtained from studies; (v) a variety of non-health and non-clinical sources reflecting lifestyle and habits (e.g., local weather, air quality level, buying habits, patient activity on social media); and (vi) hospital processes [6,12]. In particular, advancements in personal health monitoring devices in the form of mobile applications or built-in sensors can actively monitor a user’s vital health parameters such as ECG, BP, heart rate, and sugar level, which reduces the potential errors of data recording [13].

#### 2.1.2. Technology for DT Implementation

DTs are realized using a diverse set of technologies, as shown in Figure 1: Digital health devices/sensors directly collect data from the patient and/or the environment, then transmit and store it in the IoT cloud in real-time. Big data analytics and AI extract meaningful information from an extensive amount of data and enable the visualization of a virtual twin of the patient at a different stage of the disease [9,14]. IoT, cloud computing, AI, simulation, visualization tools, and machine learning models are all necessary elements to build DTs. Advancements within these technologies and additional technological domains like virtual reality (VR) and supercomputers are making the use of DTs in healthcare increasingly attractive. 

***Internet of Things:*** Developments in sensor technologies and wireless networks are contributing to the advancement in applications in the human DT concept by increasing the amount and the accuracy of collected real-world and real-time data and sharing capabilities. Improved IoT sensors and devices refer to internet-connected sensors and devices that can be embedded in everyday objects or attached to the human body like wearables. These objects can collect, send, and receive data about the physical object and/or its surrounding environment. The result is an improvement in establishing and maintaining more comprehensive simulations of physical twins, their functionality, and the changes they undergo over time [15]. 

***Cloud Storage and Computing:*** DTs require an extensive amount of data, and therefore high computing power is needed, for example, to allow clinicians to extract real-time information on patients. However, required data storage and computing capabilities may exceed the ones currently allocated in healthcare centers. Thus, many healthcare centers outsource their healthcare data and monitoring services to powerful cloud servers. When coupled with DTs, these cloud storage and computing technologies provided by servers can alleviate the computing pressure of DTs from healthcare centers [16]. In this context, the cloud server is employed for storing healthcare data and offering efficient query services, where the DT is used for building a digital representation of patients and leverages the query services to monitor the health states of patients. However, since patient data is classified as sensitive healthcare information, the potential leakage and vulnerability may cause privacy issues or even intentional catastrophic losses to patients (if their data are hacked). To enhance data security, healthcare centers usually encrypt the information before outsourcing it to the cloud [16]. 

***Artificial Intelligence:*** DT applications draw on AI technologies, machine learning, and software analytics to provide a real-time digital representation of the physical object. Currently, the DTs of AI-based human biological systems or organs help to diagnose current medical conditions and predict possible future health problems based on aggregated data and medical histories [17]. Moreover, AI assists the design of DTs of organs using their physiological data to output a 3D image. For instance, Siemens Healthineers developed a DT model by exploiting a massive database containing more than 250 million annotated images, reports, and operational data. The AI-based DT model enables digital heart design based on patient data with the same patient conditions (size, ejection fraction, muscle contraction, etc.) [1]. 

***Virtual Reality:*** To enable lifelike simulations, DTs can be paired with Virtual Reality technology in the future. This will help clinicians practice complex procedures and benefit medical education, allowing them to practice treatments and procedures on virtual patients before applying them to real ones [18]. For example, Pediatric cardiologists at Lucile Packard Children’s Hospital Stanford are already using VR technology as a teaching tool to explain complex congenital heart defects, visualizing a general model of a 3D beating heart [19].

### 2.2. Current Applications of DTs in Healthcare

#### 2.2.1. Precision Medicine and Support to Medical Decision Making

DT applications in healthcare can contribute to the broad trend of precision medicine to maximize the efficiency and efficacy of the healthcare system by shifting from current clinical practice with ‘one-size-fits-all’ treatments to take inter-individual variability into greater account [12]. ‘‘Precision medicine’’ (more generally referred to as ‘‘personalized medicine’’) is an emerging approach for disease treatment and prevention surrounding the use of new diagnostics and therapeutics targeted to the needs of a patient based on their own genetic, biomarker, phenotypic, physical or psychosocial characteristics. The aim is to deliver the right treatments, at the right time, to the right person [1]. However, most current healthcare systems are not fully able to provide personalized treatment for diseases having multi-stage diagnosis and treatment processes and high variability in terms of disease characteristics (like in the case of cancer treatment) [2]. One of the most important barriers to precision medicine is that patients with the same disease do not adequately respond to the same treatment. This is primarily due to the wide gap between the complexity of the focal condition, which may involve altered interactions between thousands of genes that alter across patients with the same diagnosis (thus identifying multiple diseases behind the same diagnosis), and modern healthcare, in which diagnostics often relies on a growing but still a relatively small number of biomarkers of limited sensitivity or specificity [10,20]. In order to address these limitations, DTs may help in creating a human model defined by all the structural, physical, biological, and historical characteristics of an individual that can be matched with thousands or millions of comparable data from other individuals, facilitating the search and identification of interesting genetic characteristics. Therefore, DTs may ease the prediction of an illness by analyzing the real twin’s personal history and the current context, such as location, time, and activity [1,12]. Furthermore, DTs may simulate the impact of a treatment on these patients and provide decision support to physicians and other healthcare professionals such as hospital pharmacists.

A comprehensive framework for DTs-based personalized medicine is presented in Figure 2. First, the model considers an individual patient showing a local sign of disease. Then, unlimited copies of patient DTs are constructed based on the high-performance computational integration of thousands of disease-relevant variables. Each virtual twin is treated with various drugs as a potential treatment for the disease, and one computational drug treatment results positively for the patient. As the final step, the drug with the best results is selected for the patient treatment. 

Additionally, for critical diseases like Multiple Sclerosis (MS) or cases like trauma or elderly management, the demand for personalized medicine is rapidly increasing since timely and precise intervention is critical. Although there has been no clear DT implementation yet, researchers have started to propose reference frameworks for their further application. For instance, Y. Liu et al. [4] presented a framework of the CloudDTH (cloud healthcare system based on DT healthcare) for elderly healthcare management and its application scenario for elderly patients. The proposed framework aims to solve the problem of real-time supervision and accuracy of crisis warnings for the elderly in healthcare services by DT, cloud computing, health IoT, and big data.

Although DTs’ use cases are still limited in scope in precision medicine, some DTs of organs (e.g., heart) or parts of the human body have already been developed and used as prototypes or pilots [1]. One precursor in the organ DTs’ environment is Dassault Systèmes’s Living Heart project, which is the first functioning computer model of the complete heart, taking into consideration all aspects of functionality, including blood flow, mechanics, and electrical impulses. Functioning Living Heart has been developed and is now available to anyone worldwide. It is used for designing new medical devices, analyzing drug safety, designing personalized surgical treatments, and is also used in biomedical education [1,21]. Some of the current applications of DTs in healthcare, particularly in precision medicine, are summarized in Table 1.

#### 2.2.2. Clinical Trials Design

Beyond their application for supporting diagnosis and treatment in healthcare, DTs might also be useful in the development phase of new treatments, particularly in the conduction of clinical trials. Approximately 80% of clinical studies face delays in the enrollment phase, and 20% of trials fail to meet overall enrollment goals [11]. The problems occurring in the enrollment stage (e.g., finding participants who fit the criteria and are willing and able to participate), coupled with the trend of personalized medicine of identifying smaller target populations, make clinical trials increasingly expensive, time-consuming, and inefficient. DTs may allow the creation of unlimited copies of an actual patient and treat them computationally with a large variety of drug combinations that could act as the control group. This way, DTs of real patients could be used to test early-stage drugs to accelerate clinical research, minimize their hazardous impact and reduce the number of expensive trials required to approve new therapies. 

The current use of DTs in clinical trial support is, however, very limited. Some studies show that DTs are promising for addressing the main challenges of clinical trials, such as designing smaller trials with higher statistical power or recovering power in ongoing trials affected by low enrollment or high dropout rates. In the short term, DTs are expected to contribute to randomized controlled trials to improve power and efficiency without introducing bias [26].

At present, UnlearnAI, a leading company in the field, is working with DTs to accelerate clinical research in Alzheimer’s Disease and Multiple Sclerosis.

***Addressing Placebos and Dummy Drugs in Clinical Trials.*** The control group in comparative clinical trials sometimes generates ethical issues when the treatment is potentially lifesaving (and the standard of care/placebo is not) or if there are important differences in the treatment’s characteristics (e.g., safety issues, invasive procedures compared to non-invasive ones, etc.). DTs can replace placebo (or standard-of-care) patients and simulate the evolution of health states based on patients’ characteristics, providing a representative view of an intervention’s impact on the virtual twin. The DT will, therefore, create a synthetic control group.

#### 2.2.3. Optimizing Hospital Operations

Another potential DT application in healthcare is the optimization of hospital operations and management. Large companies such as GE Healthcare and Siemens Healthlineers have already developed DTs and are currently tailoring their DT services for hospitals to respond to challenges such as growing patient demand, increasing clinical complexity, aging infrastructure, lack of space, increasing waiting times, and rapid advances in medical technology requiring additional equipment implementation [1]. Using DTs, different possible solutions can be tested in virtual environments before scheduling and implementation in the real setting [4] (e.g., bed planning, staff schedules, surgical simulation, and virtual drug experiments). For instance, GE Healthcare developed the Capacity Command Center to build DTs of patient pathways in Johns Hopkins Hospital in Baltimore. By applying simulations and analytics, the hospital can predict the patient activity and plans capacity according to demand, thus significantly improving patient service, safety, experience, and activity volume.

The final aim of DT application in hospital management is to help hospitals, other healthcare organizations, and policymakers to manage and coordinate patient care initiatives from a social and population perspective [1]. For instance, in extreme cases such as pandemics, hospital management can simulate different possible conditions (and their potential solutions in virtual environments before implementing them in the physical space. Finally, DT application in hospital operations will allow hospitals and other institutions to timely allocate their resources to increase efficiency, save cost and avoid predictable crises.

## 3. Open Issues and Challenges

In addition to the promising opportunities, DTs applications have a number of challenges and concerns that may hinder the full development of their potential. Since DTs combine various emerging technologies such as AI, the Internet of Things, big data, and robotics, each component brings its own socio-ethical issues to the implementation stage [27]. Additionally, since the convergence of these technologies and the DT concept in healthcare are in most cases still in their early development phase, technical limitations arise, and include everything from data collection to software design. Finally, DTs can show great performance in a short timeframe, but their predictive capability alone might not be considered sufficient for therapy selection and preventive care. 

### 3.1. Ethical Issues and Policy Considerations

***Security and privacy.*** Effective implementation of DTs in healthcare requires wide patient data collection and storage. However, the access and the integration of these sensitive patient data, including biological, physical, and lifestyle information over time by healthcare organizations or insurance companies, raise ethical questions, where the confidentiality and security of information remains paramount [12]. For instance, insurance companies may make precise distinctions for premiums based on new data points processed through DTs (e.g., physical activity, eating habits), especially on people whose health data suggests an impending negative event, making care more difficult to access at the time when it is most critically important to receive (a potential misuse of DTs) [6]. Additionally, some experts have raised concerns about the cybersecurity of DT databases, where the risk of a cyber-attack cannot be denied [27].

To address these issues, new regulations like The EU General Data Protection Regulation (GDPR) impose new legal requirements such as the right to withdraw consent and the right to be forgotten. Owners of DTs solutions should carefully monitor these requirements since new rules can also apply to historical data and safety backups [12].

***Data bias.*** DTs require a data model built on a balanced dataset where any individual’s data can be compared. However, at present, many healthcare-related datasets incorporate racial, gender, or other demographic sources of bias (e.g., white men are more represented). Using these datasets to build human DTs without any correction would intensify the already existing bias and finally result in a suboptimal recommendation system for any patient who does not fit the typical demographic profile of the dataset [6].

***Good gene pool.*** DTs can reveal what type of genetic profile tends to perform better in terms of survival and health. The presence of this information can raise concerns about political decisions based on convictions about what constitutes a good gene pool [27]. Extreme consequences of defining “good genes pools” could include the pressure to select individuals based on their genetic profile for specific jobs or even to generate the willingness to drive the selection processes of embryos based on their genetic profile (eugenics).

***Accessibility to the technology.*** As a new application in healthcare, there is not yet a clear business model on how DTs will reach patients. In case DTs-based treatments or preventions are not accessible to everyone or are not covered by health insurance, their use will widen an already existing socio-economical gap by providing access to knowledge and expertise to patients rich enough to afford the treatments themselves or whose system is not willing or able to pay for DTs applications [27].

### 3.2. Technical Limitations

***Data collection and management.*** Human DTs require deep and detailed datasets and new Electronic Health Records (EHR) designs which will foster data mining and the automated collection of clean data. Currently, an important roadblock for Human DTs is that electronic health records and healthcare information systems are highly heterogeneous and difficult to operate [12]. Moreover, information is often in an unstructured format, and its extraction requires either manual work or further implementation of automation through natural language processing technologies. 

The quality of the data supplied also plays a vital role. Although sensors can efficiently collect data and transfer it to Human DTs, hospital data collection processes can be more expensive and time-consuming [17]. Currently, most of the data from individuals is collected through blood tests, imaging systems, and health scans. The hospital data collection processes thus create a burden on DT processes. For example, it is not easy to achieve excellent image quality in computerized tomography (CT) scans of heart patients, and the output generally depends on the expertise of radiology staff; this is especially true in the case of less experienced radiology staff. Experts in the field indicate that the next big milestones in DTs will not be related to advancements in AI research but will deal with fixing the problems with small-scale, messy data in healthcare [12].

***Process and interface design.*** Although DT applications have been defined as fully autonomous processes, there is a crucial need for interdisciplinary knowledge (e.g., biomedical, mathematics, bioengineering, and computer science) and people’s experience due to the complex nature of human beings [17]. Moreover, DT software designers should work on a user-friendly interface for DTs to facilitate communication among DTs software, patients, and physicians (e.g., to discuss optimal treatment based on informed consent). However, experts in the field note a lack of user-friendly software for DT applications in healthcare [21].

### 3.3. Social Barriers

***Mistrust on decision points.*** In general, physicians still lack trust in decisions derived from algorithms and big data since these predictions are generally not matched with a plausible or transparent explanation. Recent research exploring AI systems integration in the hospital environment shows that many physicians are skeptical of AI given the high risk tied to potential misdiagnosis and improper treatment [28].

***Fear of clinician replacement.*** The fear of clinician replacement may arise with the broader use of DTs in clinical tasks [12]. In some cases, DTs may outperform clinicians, since it is not likely that a clinician could process all data from the patient and provide a solution in a short visit. However, considering the mistrust of decision points of DTs and the current state of DT applications in healthcare, in the future, DTs will aim to adapt to the needs and workflows of clinicians and enhance their ability to efficiently consider the whole set of available information when making decisions.

## 4. Discussion

DTs may deliver significant value to healthcare processes. In particular, their contribution may move precision medicine (including prevention) to a higher level. They could solve growing issues in the development phase of new technologies (synthetic arms in clinical trials) and help hospital managers optimize operations and policymakers anticipate new health policies’ consequences. However, while DTs application to systems and organizations is already well established in other industries (automotive, space engineering, etc.), when a human being and their life and rights are involved, the use of DTs needs to be framed in a much more complex environment. Many barriers, indeed, prevent DTs from currently developing their full potential in healthcare. Some are technical (e.g., data computing capabilities), some are cultural-technical (lack of interoperability of systems where patient data are loaded and stored), and some are related to ethical and social concerns. Technical barriers are common to most applications of DTs, also outside healthcare, and will be gradually overcome according to technological advancements. Cultural-technical barriers are more resistant since they entail a cultural resistance to sharing data and consequently losing the exclusivity of their use. When data are collected through different methods by different organizations, a cultural work is needed to create the willingness to standardize, make uniform and share the collected information. We recommend that at the level of each country a central institution or a network is made responsible for promoting this cultural work, gradually moving data ownership towards a centralized one, which will also define access rights and sharing benefits (e.g., physicians can instantly identify a DT for their patients based on the information shared by all centers in their country). Multinational initiatives are also promising, provided that the country-level cultural work has created sufficient support. With respect to ethical and social issues, however, the debate on DTs is still in its early stages, and we encourage scholars, companies, policymakers, and practitioners to dedicate more space to the discussion of regulatory aspects, organizational effects, ethical concerns, and social implications of DTs. To start a more structured reflection, we provide a short list of questions and arguments.

### 4.1. Use of DTs in Clinical Trials

So far, the research and development cycle of new drugs and (with less structure) new medical devices has been institutionalized and is heavily based on the use of real human beings. In comparative studies, the randomization allows for proxies of the counterfactual outcome of the treatment group. The control group is selected with homogeneous characteristics compared to the treatment group since it should be a “reasonable” copy whose reaction to the “control” (placebo or standard of care) is likely to be the same that the individuals included in the treatment group would have shown if they were assigned to receive the placebo. Under this perspective, if the control group were composed of DTs of the treatment group, it would be theoretically more adherent to reality than other similar individuals, so researchers should trust the DTs-based control group more. However, researchers could instead not trust DTs since there are no real humans, and their reaction to drugs or other treatments might not be identical to the one displayed by a real person (mainly when data sources to inform the DTs are poor). In short, the use of DTs in clinical trials entails the solution to a dilemma: should we accept the variability between the real human being and its DT more or the variability between different human beings having similar baseline characteristics? The answer should be given after a gradual testing phase of DTs. A possible way is to introduce DTs for trials where there are objective difficulties in collecting participants, mainly for the control group. Comparing real and virtual controls could inform researchers about the steps required before DTs’ more extensive use in clinical trials.

### 4.2. Equity Issues 

As stated before, the current healthcare information is biased in terms of the demographic characteristics of the represented populations. In this perspective, DTs may exacerbate this bias, providing more opportunities for prevention and care for the most represented populations, leaving minorities behind. However, given the capacity of DTs to balance the lack of information, they could also be used as opportunities to over-represent minorities and conduct studies on their specific health needs with fewer resources (at regime).

### 4.3. Social Issues Related to the Affordability of DTs

DTs are expensive technologies, at least in their current early diffusion phase. Therefore, their cost constitutes a barrier for poor patients or poor systems. This barrier is not different from the one that is concerned with most technological innovations in healthcare and should be treated accordingly. Price differentiation is a possible solution that is already used for drugs and medical devices.

### 4.4. Social Issues Related to the Potential Replacement of Clinicians 

DTs are meant to be a decision support: they may complement lacking information, or they may allow clinicians to leverage an efficient information set, but in no way are able to fully replace medical decision making, at least within the paradigm within which they are currently developed. Replacing a physician in their decision-making would require a shift in the ultimate responsibility of a choice and, currently, this is not placeable on a software or a virtual copy. Therefore, even the most informative DT always requires (and we think that it will require) a human to make a final decision.

## 5. Conclusions

This paper has reviewed the current applications of DTs in healthcare, showing that despite the impressive potential of their use, several barriers are now slowing down their diffusion. The role that DTs could play to support medical decision-making is currently limited mostly by computational, ethical and cultural concerns, suggesting that technological capacity and cultural acceptance should evolve together and develop an adequate level of trust. We recommend the inclusion of DTs in education programs and national plans for pilot-diffusion projects of DTs to avoid entropy and miscoordination. At the same time, excessive confidence in technology alone should be avoided, since DTs represent a decision support but are not meant (nor they are able) to replace clinical decision-making. Moreover, the use of DTs could be very helpful in solving relevant issues in evidence generation, where real data are difficult to obtain because of small numbers of rare diseases or ethical barriers. However, the methodology for including DTs in clinical trials is lagging and there is not yet homogeneous support from the scientific community, yet. In this perspective, we encourage the development of ad-hoc methods, where required, to exploit the potential of DTs in evidence generation.

## Figures and Tables

**Figure 1 jpm-12-01255-f001:**
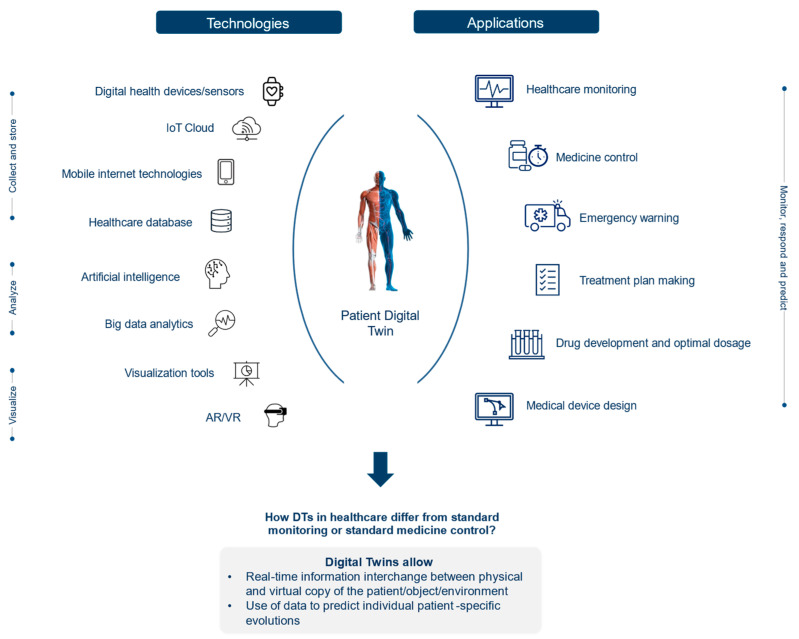
DT technologies: Working scheme.

**Figure 2 jpm-12-01255-f002:**
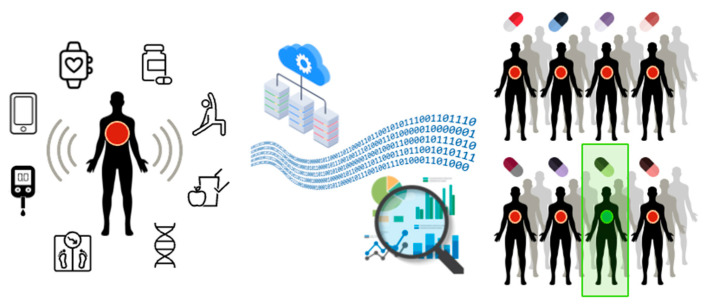
The DT concept for personalized medicine.

**Table 1 jpm-12-01255-t001:** Current DT Application in Precision Medicine and Medical Decision-Making Support.

Target Organ/Disease	Reference (Company, Journal etc.)	Description
Heart [1]	Living Heart Project, Dassault Systèmes	The Living Heart Project is the first DT organ considering all aspects of the heart’s functionality, such as blood flow, mechanics, and electrical impulses. The 3D model of the organ has built with a 2D scan of the heart. The Living Heart Model on the 3DEXPERIENCE platform can be used to create new ways to design and test new devices and drug treatments. For instance, physicians can run hypothetical scenarios like adding a pacemaker or reversing the heart chambers to predict the outcome of treatment on the patient.
Heart [12]	CardioInsight, Medtronic	The CardioInsight Noninvasive 3D Mapping System collects chest electrocardiogram (ECG) signals and combines these signals with computerized tomography (CT) scan data to produce and display simultaneous 3-D cardiac maps. The mapping system enables physicians to characterize abnormal rhythms of the heart through a personalized heart model.
Heart [1]	Siemens Healthineers	Another heart DT has been developed by Siemens Healthineers and used for research purposes by Cardiologists of the Heidelberg University Hospital (HUH) in Germany. Although the first study is still in the data evaluation process, preliminary results are promising. Siemens Healthineers developed the DT model by exploiting a massive database containing more than 250 million annotated images, reports, and operational data. The AI-based DT model enables digital heart design based on patient data with the same conditions of the given patient (size, ejection fraction, and muscle contraction).
Brain [17]	Blue Brain Project, EPFL and Hewlett Packard Enterprise	Hewlett Packard Enterprise, partnering with Ecole Polytechnique Fédérale de Lausannes (EPFL), builds a DT of brain called the Blue Brain Project. The project is one of the sub-projects of the Human Brain Project and aims to build biologically detailed digital reconstructions (computer models) and simulations of the mouse brain. In 2018, researchers published the first 3D cell atlas for the entire mouse brain [22].
Human air-way system [1]	Oklahoma State University’s Computational Biofluidics and Biomechanics Laboratory	Researchers developed a prototype of human DT, named ‘‘virtual human V1.0”, with the high-resolution human respiratory system covering the entire conducting and respiratory zones, lung lobes, and body shell. The project aims to study and increase the success rate of cancer-destroying drugs in targeting tumor-only locations.
Brain aneurysm and surrounding blood vessels [1]	Sim&Cure	Sim&Cure developed a DT to treat aneurysms, which are enlarged blood vessels that can result in clots or strokes. DT of the aneurysm and the surrounding blood vessels (represented by a 3D model) allow brain surgeons to run simulations and understand the interactive relationship between the implant and the aneurysm. Although preliminary trials have shown promising results, further evaluation is required.
Multiple Sclerosis (MS) [9]	Frontiers in Immunology (journal)	Multiple sclerosis, also called the ‘disease of a thousand faces’, has high complexity, multidimensionality, and heterogeneity in disease progression and treatment options among patients. This results in extensive data to study the disease. Human DTs are promising in the case of precision medicine for people with MS (pwMS), allowing healthcare professionals to handle this big data, monitor the patient effectively, and provide more personalized treatment and care.
Viral Infection [23]	Science (journal)	Human DTs can predict the viral infection or immune response of a patient infected with a virus by integrating known human physiology and immunology with population and individual clinical data into AI-based models.
Trauma Management [24]	Journal of Medical Systems (journal)	Trauma management is highly critical among time-dependent pathologies. DTs can participate from the pre-hospital phase, where the physician provides the patient first aid and transfers them to the hospital emergency department, to the operative phase, where the trauma team assists the patient in hospital emergency. Although there is no real implementation yet, a system prototype has been developed.
Diabetes [25]	Diabetes (journal)	Human DT can also participate in diabetes management. California-based start-up Twin Health has applied DTs by modeling patient metabolism. The DT model tracks nutrition, sleep, and step changes and monitors patients’ blood sugar levels, liver function, weight, and more. Ongoing clinical trials show that daily precision nutrition guidance based on a continuous glucose monitoring system (CGM), food intake data, and machine learning algorithms can benefit patients with type 2 diabetes.

## Data Availability

Not applicable.

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
