# Peer review of "Digital Twins in Healthcare: Is It the Beginning of a New Era of Evidence-Based Medicine? A Critical Review"

_jpm, 2022, doi:10.3390/jpm12081255_

Round 1

Reviewer 1 Report

The article, presenting a review of the current and potential use of Digital twins in medicine and healthcare, presents usefully structured and analysed information on this novel topic.

I have the following remarks to make:

1. a Conclusions section is missing

2. in table 1:

             -  for Sim&Cure - please rephrase the description for a better understanding

Reviewer 2 Report

I found your paper very informative and well-researched. I also found the title misleading. Nothing is found in the paper regarding Medical Decision-Making except in the title. Neither an answer to the question: Is it the end of traditional evidence-based medicine? is provided. Your paper needs a new title, one that really describes what you paper presents and discusses.

Your paper reads like a literature review of the state of Digital Twin (DT) technology in health care environments. Therefore think about giving it a title that match the content. That being said. I suggest you start by discussing the definitions of Digital Twin that match your literature. As you mentioned, there are several ways in which digital twins are defined and enacted. To begin, from the perspective of the use of digital twins in managerial decision-making, a digital twin is a way to manage assets by creating a digital copy of the asset and using IoT sensors to enable a continue communication between the asset and the digital copy. Therefore, a digital twin of a patient, seems to have to be explained, especially when it comes to the using of IoT sensors. You can add physical sensors into a physical asset, RFID tags are commonly used for this purpose. It is less clear how you can add IoT sensors to patients.

I looked at your sources, especially those used when explaining Figure 1, and still cannot fully understand the role the physical patients play in it. How it is different from what hospitals generally do when collecting data from patients, i.e., blood, urine samples, x-rays, CT scans, etc. which deals with the actual patient, but not necessarily becomes a digital twin? You clearly say in page 4, lines 135/136 "real-time data connection and exchange between physical and virtual objects should be executed through health IoT and mobile internet technologies". But that is not clear in your DT model.

I see a misinterpretation of digital models for some aspects of a patient's data as a digital twin. Health care uses several independent devices to monitor patients' vital signs. These provide information to health care providers on which they engage in decision-making, but these are not digital twins of a patient, only monitor devices.

The paper will benefit greatly by actually including a section on Medical Decision-Making, its strengths and weaknesses, and propose a way to assess if a digital twin can actually help overcome some of the weaknesses that currently are inherent on their decision-making processes. This can be done by looking at the benefits offered by the studies you reviewed. 

Also, there are (minor) issues with the language of the paper, 

In line 354, page 10, you use a word that does not exist in English beyond a literary work: herselves, must be themselves.

In line 386, you say: "many physician is skeptical," it should be: "many physicians are skeptical"
